# SPATIAL AND TEMPORAL VARIABILITY OF WATER-FILLED CREVASSE HYDROLOGIC STATES ALONG THE SHEAR MARGINS OF JAKOBSHAVN ISBRAE, GREENLAND

<sup>1</sup>Casey A. Joseph, and <sup>1</sup>Derrick J. Lampkin

1 Department of Atmospheric and Oceanic Sciences, University of Maryland, College Park, MD 20742, USA

Correspondence to: Casey A. Joseph (cajoseph@terpmail.umd.edu)

#### Abstract

The impact of melt water injection into ice streams over the Greenland Ice Sheet is not well understood. Water-filled crevasses along the shear margins of Jakobshavn Isbræ are known to fill and drain, resulting in weakening of the shear margins due to reduced basal friction. Seasonal variability in the hydrologic dynamics of these features has not been

- quantified. In this work, we characterize the spatial and temporal variability in the hydrological state (filled or drained) of these water-filled crevasse systems. A fusion of multi-sensor optical satellite imagery was used to examine hydrologic states from 2000 to 2015. The monthly distribution of crevasse systems observed as water filled is unimodal with peak number of filled days during the month of July at 329 days, while May has the least at 15. Over the study period the occurrence of
- drainage within a given season increases. Inter-seasonal drain frequencies over these systems ranged from 0 to 5. The frequency of multi-drainage events are correlated with warmer seasons and large strain rates. Over the study period, summer temperatures averaged from -1 and 2 °C and tensile strain rates have increased to as high as  $\sim 1.2$  s<sup>-1</sup>. Intermittent melt water input during hydrofracture drainage responsible for transporting surface water to the bed is largely facilitated by high local tensile stresses. Drainage due to fracture propagation may be increasingly modulated by ocean-induced calving dynamics for
- the lower elevation ponds. Water-filled crevasses could expand in extent and volume as temperatures increase resulting in regional amplification of ice mass flux into the ice stream system.

#### **1 Motivation and Prior Work**

The Greenland Ice Sheet (GrIS) has experienced considerable mass loss over the last few decades (Krabill et al., 2004; Joughin et al., 2004; Alley et al., 2005a, 2005b; Luthcke et al., 2006; Hanna et al., 2008) resulting in negative mass balance

and substantive contribution to sea level rise (Rignot et al., 2008; van den Broeke et al., 2009; Shepherd et al., 2012). Commensurate with these changes has been the documented impact of surface meltwater on ice sheet velocity during the summer within the ablation zone (Zwally et al., 2002; Joughin et al., 2008; van de Wal et al., 2008; Shepherd et al., 2009; Bartholomew et al., 2010; Sundal et al., 2011; Palmer et al., 2011; Hoffman et al., 2011), via supraglacial lakes, channels,

and moulins largely beyond regions of fast flow (Echelmeyer et. al., 1991; Box and Ski, 2007; McMillan et al., 2007; Sneed and Hamilton, 2007; Das et al., 2008, Sundal et al, 2009; Lampkin, 2011; Selmes et al., 2011; Tedesco and Steiner, 2011; Howat et al., 2013; Koenig et al., 2015). The presence of ponded water within regions of fast flow has received little attention. Lampkin et al. (2013) evaluated the spatial and temporal variability of water-filled crevasse filling and drainage

- dynamics during the 2007 melt season within the shear margins of Jakobshavn Isbræ. Crevasses at elevations less than ~500 m start to fill around June 6 with a total area of ~0.15 km<sup>2</sup>. A peak total area of ~1.8 km<sup>2</sup> was reached in early July with most groups still maintaining some water on August 9, 2007. Water-filled crevasse systems filled and drained at rates as large as 0.03 km<sup>2</sup> d<sup>-1</sup> and 0.012 km<sup>2</sup> d<sup>-1</sup> respectively (Lampkin et al., 2013). These features have the capacity to inject substantial volumes of water into the shear margins equivalent to the largest supraglacial lakes found outside of the ice stream (Lampkin
- et al., 2013). We do not understand how these structures have changed over time, their specific impact on regional ice dynamics, and the mechanisms through which melt water from these features can be delivered to the bedrock.

# 1.1 Objectives

This investigation performs the most comprehensive assessment of the spatial and temporal variability of water-filled crevasses along the shear margins of Jakobshavn Isbrae. We seek to characterize variability in drainage dynamics over

- water-filled crevasse systems at annual and interannual time scales using a fusion of multi-sensor data from several optical satellite systems acquired over a 16 year period from 2000 to 2015. We restrict our assessment to characterizing the 'hydrologic state' (filled or drained) of these water-filled crevasse systems over the analysis period. We do not quantify changes in the volume or areal extent of these systems. We characterize temporal patterns in the drain state of water-filled crevasse systems responsible for hydrologic weakening of Jakobshavn Isbrae. Lastly, we explore first-order controls on
- observed drainage behavior. This work provides an important benchmark from which future changes in this component of supraglacial hydrology and the role of meltwater in fast flowing ice streams. This work has implications for understanding processes driving mass discharge from marine-terminating outlet glaciers throughout GrIS.

#### 2 Data

Multi-sensor satellite data operating over the visible part of the EM spectrum are used to characterize the spatio-temporal variability of water-filled crevasses hydrologic states. We examine spatial and temporal variability of surface strain fields based on satellite-derived measured velocity data. Additionally, near surface air temperatures to evaluate meteorological conditions associated with melt production and runoff that can influence the hydrologic state of the water-filled crevasse systems.

5

# 2.1 Satellite Imagery

Satellite imagery was acquired from several optical satellite systems spanning a range in performance capacity. Cloud-free images from seven imaging systems are used to quantify the hydrologic state of each water-filled crevasses system. The presence of ponded water is easily identified in imagery acquired over the visible part of the electromagnetic spectrum resulting from the propensity for water to absorb incoming solar radiation more effectively than the surrounding ice and firm (Lampkin and VanderBerg, 2011). Data from Landsat-7 ETM+, Landsat-8 OLI, Quickbird-2, Worldview-1/2, EO-1 ALI, SPOT-5, ASTER, and Google Earth are used in this analysis to offset the relative performance limitations inherent to the use of data from a single system. The most effective performance parameter optimized through the use of data from several sensors is the temporal resolution. The number of cloud-free images varies for each satellite system, resulting in a non-

10 periodic sampling interval. The overall temporal resolution is improved though the sampling rate is inconsistent. For more details on imagery sources and image resolutions see Table 1.

#### 2.2 Surface Temperature Data

Near surface temperature data were acquired from the Greenland Climate Network (GC-Net) near Jakobshavn Isbrae from the Cooperative Institute for Research in Environmental Sciences (CIRES) (Steffen et. al., 1996). Hourly, 2 m surface temperatures sampled at the JAR 1 and Swiss Camp GC-NET stations are used to create a composite daily average temperature. This time series is used to evaluate patterns in the filling and drainage variability of water-filled crevasse systems.

# 20 2.3 Velocity Data

Velocity data used in this analysis were derived from the National Science and Ice Data Center (NSIDC) MEaSUREs data archive on Greenland Ice Velocity: Select Glaciers InSAR (release v1.1) (Joughin et al., 2016) (<u>http://nsidc.org/data/nsidc-0481/</u>). Surface velocity fields are derived from TerraSAR-X (TSX) image pairs collected from 2009 to 2016 based on speckle tracking and interferometric techniques (Joughin et al., 2010, 2016). Available transverse and longitudinal

25 component surface velocity grids were acquired from this archive over the Jakobshavn Isbrae study area from 2009 to 2015 during the months of May through August. Nominal spatial resolution of these grids are at 100 m.

#### **3 Methods**

#### 3.1 Determination of Hydrologic State of Water-filled Crevasses

The hydrologic state of water-filled crevasse systems are quantified through visual interpretation of visible imagery. The occurrence or presence of water within the previously identified water-filled crevasse groups indicate a "filled" hydrologic state. A crevasse group is assumed to remain filled until a subsequent image indicates that a particular group is devoid of water. We assume a given crevasse group remains water-filled during intervening periods when conditions prevent direct observation. This scenario can occur when an initial cloud-free image displays water in a system and is followed by a period of cloud-covered or lack of available images. After such a period, if the subsequent image no longer shows water present then we assume drainage occurred during the intervening interval. In general, when a crevasse was observed to be filled, we

assumed the crevasse was filled until we either observed the crevasse to drain, or the study period for that given year ended.

#### 3.2 Derivation of Strain Rate

The surface velocity field is decomposed into components consisting of vector (u) oriented in the prevailing direction of ice flow (x) within the main trough of the ice stream, and an orthogonal component (v) perpendicular to ice flow (y). Horizontal

strain rates ( $\dot{\epsilon}$ ) are estimated from measured surface velocity through differentiation of component velocity grids where the strain rate tensor is given by:

$$\begin{bmatrix} \dot{\varepsilon}_x & \dot{\varepsilon}_{xy} \\ \dot{\varepsilon}_{yx} & \dot{\varepsilon}_y \end{bmatrix} = \begin{bmatrix} \frac{\partial u}{\partial x} & \frac{1}{2} \left( \frac{\partial u}{\partial y} + \frac{\partial v}{\partial x} \right) \\ \frac{1}{2} \left( \frac{\partial v}{\partial x} + \frac{\partial u}{\partial y} \right) & \frac{\partial v}{\partial y} \end{bmatrix}$$
(1)

Strain rate component fields are used to calculate the magnitude in the principle strain axis in the horizontal plane, where the magnitude of minimum  $(\dot{\epsilon}_1)$  and maximum  $(\dot{\epsilon}_3)$  tensile strains are given by:

$$\dot{\varepsilon}_{1} = \frac{1}{2} (\dot{\varepsilon}_{x} + \dot{\varepsilon}_{y}) - \sqrt{\left[\frac{1}{4} (\dot{\varepsilon}_{x} - \dot{\varepsilon}_{y})^{2} + \dot{\varepsilon}_{xy}^{2}\right]}$$
(2)  
$$\dot{\varepsilon}_{3} = \frac{1}{2} (\dot{\varepsilon}_{x} + \dot{\varepsilon}_{y}) + \sqrt{\left[\frac{1}{4} (\dot{\varepsilon}_{x} - \dot{\varepsilon}_{y})^{2} + \dot{\varepsilon}_{xy}^{2}\right]}$$
(3)

In this analysis, we are specifically interested in the  $\dot{\varepsilon}_3$  field as we want to examine the spatial and temporal variability in the tensile strain field which controls fracture propagation within the shear margins. Given this, we do not compute the angle between  $\dot{\varepsilon}_1$  and  $\dot{\varepsilon}_3$ .

#### 4 Results

#### 4.1 Spatial and Temporal Variability of Hydrologic State

The availability of cloud-free images varies across the study period with some years having more samples than others. The number of total samples per season increases with time (Figure 2a). The maximum scenes available for any given month in the study period was 87 in June 2010. There were 17 months in the study period that cloud-free imagery was not available (Figure 2a). The total amount of imagery across all seven CV groups ranged from 163 to 228 scenes. The total number of cloud-free images varied for each crevasse system throughout the study period. The month of May maintained the least amount of clear scenes at 55, while July has the most at 433 scenes (Figure 2b). The monthly distribution of filled crevasses over the 16 year study period is unimodal with a peak in the number of filled days during the month of July at 329 days,

10 while May has the least at 15 days (Figure 3). Specifically, CV2 has the largest number of filled days among all the systems with 169 days, and CV7 has the least at 99 days.

#### 4.2 Drain Frequency and Temperature

Throughout the study period water-filled crevasse systems were observed to fill and drain throughout each season (Figure 3).
Some systems were observed to refill and drain multiple times during a season. Over the 16 year study period, there were 9 seasons where at least one crevasse group demonstrated a multi-drain event. For the 2011 season 13 drainage events were observed, with CV1 draining 5 times, both of which were maximums for the study. In 2003 we only observed 5 drainage events, indicating that 2 crevasse groups did not have water observed. Temporal patterns in seasonal hydrologic state over all crevasse groups was also examined. Time series of cumulative days over which crevasse systems were filled with water

- ( $\Delta_{\text{fill}}$ ) demonstrate five distinctive multi-year patterns (Figure 4a). These patterns generally range between 3-4 years and were consistent across all groups. The mean number of filled days per month were evaluated (Figure 4b), and varied across the seven crevasse groups. All seven crevasse groups maintain the same range. CV1 had the lowest mean at 11.21 days, while CV2 had the maximum mean at 16.75 filled days per month. Mean filled days across all groups were within 1 $\sigma$  of each other indicating that most groups do not demonstrate significant differences in the duration of filled states.

## 4.3 Relationships between Drain Frequency and Near-Surface Temperature

Relationships between drain frequency and near-surface atmospheric temperature were explored (Figure 5). Summer temperatures averaged from -1 and 2 °C for each season. From 2000 to 2006, average summer temperatures varied only between 1 to 0 °C. Commensurately, water-filled crevasse groups only demonstrate single drainage events. After 2006 temperatures increase to as high as 2 °C and vary over a larger range as high as ~4 °C difference between successive seasons.

During this period, the number of drainage occurrences increase per season across all crevasse groups with some variation.

#### 4.4 Relationships between Strain Rate and Drainage

Seasonal variations in maximum tensile strain rates can amplify fracture propagation, which may be correlated with drainage dynamics within the water-filled crevasse systems. There are only 2 multi-drain events over all crevasse groups before 2009

- 5 (Figure 5). From 2009-2015 there are 18 multi-drain events across all crevasse groups. Unfortunately, we are not able to correlate changes in strain rate with observed drain occurrence because we did not always have velocity data available for each crevasse group over every season. Generally, strain rates increased over most groups from 2009 to 2015. CV4, and 5 show the largest increase in strain during this period of ~ 1.2 s<sup>-1</sup> and 0.9 s<sup>-1</sup> respectively. The range in tensile strain rate magnitudes varies across the water-filled crevasse groups. CV1, 4 and 5 demonstrate the largest magnitudes, while CV3, 6,
- and 7 are the lowest. Specifically, the CV1 group experienced an increase in multi-drain events in 2010 and 2011 with a commensurate increase in tensile strain, which reaches a peak of ~1.2 s<sup>-1</sup> with no further changes afterwards. Interestingly, CV2 has only one season where it demonstrates multiple drainage events (2011). From 2009 to 2015, strain rates over CV2 increase slightly from 2009 to 2010 but decrease by 0.1 s<sup>-1</sup> in 2011. After 2011, strain rates rise dramatically but CV2 returns to draining only once per season. The CV3 group experience four multiple drainage events throughout the study period.
- Strain rates for CV3 were only available from 2009 to 2010 and 2012 to 2013. Strain rate values ranged from ~ 0.06 to 0.1 s<sup>-1</sup>. Over these four seasons, CV3 maintained three multiple drainage events with 2010 being the exception. CV4 shows four seasons where multiple drainage events occurred. Strain data for this group was available from 2009-2013, showing increasing strain from ~0.8 to 2 s<sup>-1</sup>. During this period, CV4 shows an increase in the occurrence of multiple drainage events. CV5 is similar and experiences an increase in multiple drainage events as well. CV6 was observed to experience four multiple.
- drain years. CV6 and 7 had limited data available from which to estimate tensile strain rates. Therefore only the 2009, 2010, 2012, and 2013 seasons are shown. Strain rates over CV6 ranged from ~0.12 to 0.18 s<sup>-1</sup>, which occurs during a period with a higher frequency in occurrence of multiple drainage events than the period before 2009. Lastly, CV7 has only one multiple drainage event throughout the entire study period during the 2009 season which does not correlate to the observed periods of strain increases.

# 4.5 Relationship between Terminus Location and Drain Occurrence

Fluctuations in local strain rates in the vicinity of each CV group could be induced by downstream calving events at the glacier terminus. We track seasonal and inter-seasonal changes in calving front location from 2002 to 2015 (Figure 6). Over this period of time, the terminus of Jakobshavn Isbræ has retreated inland substantially (Figure 6). We track the mean seasonal distance from the lowest elevation water-filled crevasses system (CV1) ( $\overline{D}_{CV1}$ ) to the terminus. From 2002 to 2013 the terminus has retreated ~ 8 km towards CV1. By 2015, the glacier front is ~ 2 km away from CV1. Inter-seasonal changes in front location ( $\Delta D_f$ ) show large variations over the analysis period. The shorter the interval over which the front locations

were observed, the smaller the magnitude in front movement. The 2004 and 2008 summers show the largest magnitude in front retreat of  $\sim 2.5$  km. Generally, the magnitude of change in frontal movement decreases over the analysis period.

Additionally, we examine relationships between front changes and the timing of drainage from each CV group over the study period (Figure 6, top panel). We document the magnitude of observed frontal change ( $\Delta D_{CV}$ ) that corresponds to

- 5 the time period when drainage was observed to occur from the water-filled crevasse groups. During the 2003 season, all CV groups drain during a period when  $1.5 

region is under tension (van der Veen, 1998, 2007). Lampkin et al. (2013) establish that local strain rates are sufficient to drive fractures through to the bed for most CV groups. The ability for water-filled fractures and crevasses to propagate through a given ice thickness to the bed is known as hydrofracture (Alley et. al., 2005b; Das et al., 2008). The water-filled crevasses examined in this analysis are in a field of closely-spaced fractures. An air-filled fracture in a field of other crevasses maintains a lower net stress intensity factor at the fracture tip, which requires a larger tensile stress to propagate 5 the fracture to the bed (van der Veen, 1998). Mean distance between fractures within the boundaries of the CV groups can range from  $\sim$ 78 m (CV7) to 110 m (CV3), which corresponds to stress intensity factors between  $\sim$ 0.5 to 0.7 (MPa m<sup>1/2</sup>) (van der Veen, 1998). These values exceed the ice fracture toughness (0.1 - 0.4 MPa  $m^{1/2}$ ) (van der Veen, 1998), but only for the case where the fractures are water-filled. A water-filled crevasse can readily penetrate to the bed because the density of water 10 is greater than ice such that if the fracture remains water-filled, the resulting hydrostatic pressure is sufficient to overcome lithostatic pressures (van der Veen, 1998, 2007). Therefore, the filling rate is the most important factor controlling fracture propagation (van der Veen, 2007). Filling rates estimated during the 2007 melt season ranged from 0.04 to 1.25 (m h<sup>-1</sup>) (Lampkin et al., 2013). Given these rates, for a tensile stress of ~ 300 kPa, a single fracture could penetrate between ~400 to 1100 m, which is equivalent to ice thickness in the vicinity of the water-filled crevasses. Melt water production and runoff

- responsible for filling crevasses are variable both intra and inter-seasonal. This would induce intermittent hydrofracture crack propagation and may not allow for a fracture to penetrate to the bed. Under these circumstances, delivery of meltwater to the bed could only be possible if local strain rates are sufficiently large to overcome the reduced stress intensity in the closely-spaced crevasse fields. Estimated strain rates during the 2007 season over the water-filled crevasse systems are sufficiently large (Lampkin et al., 2013). In this analysis maximum tensile strain rates have increased over the last 16 year
- and are correlated with an increase in multiple drainage events. The events are likely driven by both an increase in melt production and local tensile stress. This is consistent with laser altimetry estimates of surface roughness, which show substantial variability within the shear margins relative to the main trough from 2003 to 2009 (Herzfeld et al., 2014). There was no expansion of roughness within the shear margins during this period of rapid thinning (10-15 m a<sup>-1</sup>) (Herzfeld et al., 2014). Ocean-induced terminal perturbations have been implicated in the observed acceleration and thinning in the lower
- trunk of the ice stream (Holland et al., 2008; Joughin et al. 2008; Aschwanden et al., 2016). Bondzio et al. (2015) assert that ice acceleration within the main trough of Jakobshavn increases strain along the shear margins, while amplifying rheological softening. Therefore, ice fracture toughness would be reduced enough to easily facilitate fracture propagation. Clearly, the impact of terminal perturbations on strain rates in the vicinity of the water-filled crevasse groups are negligible over much of the analysis period. It is only during recent seasons that the front has reached a position such that the lower elevation systems
- (CV1, 2, and 4) are within the 10-15 km range (Joughin et al., 2012) where longitudinal coupling from calving events could be influential. This may change as the terminus of Jakobshavn continues its rapid retreat. Enhanced mass flux from Jakobshavn Isbræ over the last couple decades is driven by a combination of various factors. In particular, the impact of hydrologic weakening of the shear margins could increasingly become a major factor in both enhancing extra-marginal ice flow as well as amplifying the impact of ocean-induced terminal perturbations. Current trends and projections indicate a

warmer arctic. Under prognostic scenarios, we could expect expansion in both the distribution of ponded water in the shear margins to higher elevations and areal extent. Large volumes of water would become available for infiltration driving regional changes in ice dynamics. Hydrologic reconditioning of the shear margins could play a critical role in the future stability of not only Jakobshavn, but other outlet glaciers that maintain an active supraglacial hydrologic system.

# Acknowledgements

This work was supported under National Aeronautics and Space Administration grant NNX14AO68G. We would like to thank the anonymous reviewers for their valued time and input.

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

#### 10

Bartholomew, I., Nienow, P., Mair, D., Hubbard, A., King, M. A. and Sole, A.: Seasonal evolution of subglacial drainage and acceleration in a Greenland outlet glacier, Nat. Geosci., 3(6), 408–411, doi:10.1038/ngeo863, 2010.

Benn, D. I., Hulton, N. R. J. and Mottram, R. H.: "Calving laws", "sliding laws" and the stability of tidewater glaciers, in 15 Annals of Glaciology, vol. 46, pp. 123–130., 2007.

Bondzio, J. H., Seroussi, H., Morlighem, M., Kleiner, T., Rückamp, M., Humbert, A. and Larour, E.: Modelling the dynamic response of Jakobshavn Isbræ, West Greenland, to calving rate perturbations, Cryosph. Discuss., 9(5), 5485–5520, doi:10.5194/tcd-9-5485-2015, 2015.

#### 20

Box, J. E. and Ski, K.: Remote sounding of Greenland supraglacial melt lakes: Implications for subglacial hydraulics, J. Glaciol., 53(181), 257–265, doi:10.3189/172756507782202883, 2007.

Das, S. B., Joughin, I., Behn, M. D., Howat, I. M., King, M. A., Lizarralde, D. and Bhatia, M. P.: Fracture Propagation to the Base of the Greenland Ice Sheet During Supraglacial Lake Drainage, Science., 320(5877), 2008.

25

Echelmeyer, K., Clarke, T. S. and Harrison, W. D.: Surficial glaciology of Jakobshavn Isbrae, West Greenland: Part I. Surface Morphology, J. Glaciol., 37(127), 368–382, 1991.

Hall, D. K., Comiso, J. C., Digirolamo, N. E., Shuman, C. A., Box, J. E. and Koenig, L. S.: Variability in the surface
temperature and melt extent of the Greenland ice sheet from MODIS, Geophys. Res. Lett., 40(10), 2114–2120, doi:10.1002/grl.50240, 2013.

Hanna, E., Huybrechts, P., Steffen, K., Cappelen, J., Huff, R., Shuman, C., Irvine-Fynn, T., Wise, S. and Griffiths, M.: Increased runoff from melt from the Greenland Ice Sheet: A response to global warming, J. Clim., 21(2), 331–341, doi:10.1175/2007JCLI1964.1, 2008.

- Herzfeld, U. C., McDonald, B., Wallin, B. F., Krabill, W., Manizade, S., Sonntag, J., Mayer, H., Yearsley, W. A., Chen, P. A. and Weltman, A.: Elevation changes and dynamic provinces of Jakobshavn Isbræ, Greenland, derived using generalized spatial surface roughness from ICESat GLAS and ATM data, J. Glaciol., 60(223), 834–848, doi:10.3189/2014JoG13J129, 2014.
- Hoffman, M. J., Catania, G. A., Neumann, T. A., Andrews, L. C. and Rumrill, J. A.: Links between acceleration, melting, and supraglacial lake drainage of the western Greenland Ice Sheet, J. Geophys. Res. Earth Surf., 116(4), F04035, doi:10.1029/2010JF001934, 2011.

Holland, D. M., Thomas, R. H., de Young, B., Ribergaard, M. H. and Lyberth, B.: Acceleration of Jakobshavn Isbræ 15 triggered by warm subsurface ocean waters, Nat. Geosci., 1(10), 659–664, doi:10.1038/ngeo316, 2008.

Howat, I. M., de la Peña, S., van Angelen, J. H., Lenaerts, J. T. M. and van den Broeke, M. R.: Brief Communication "Expansion of meltwater lakes on the Greenland Ice Sheet," Cryosph., 7(1), 201–204, doi:10.5194/tc-7-201-2013, 2013.

Joughin, I., Abdalati, W. and Fahnestock, M.: Large fluctuations in speed on Greenland's Jakobshavn Isbræ glacier, Nature, 432(7017), 608–610, doi:10.1038/nature03130, 2004.

Joughin, I., Das, S. B., King, M. A., Smith, B. E., Howat, I. M. and Moon, T.: Seasonal speedup along the western flank of the Greenland Ice Sheet., Science., 320(April), 781–3, doi:10.1126/science.1153288, 2008.

Joughin, I., Smith, B. E., Howat, I. M., Scambos, T. and Moon, T.: Greenland flow variability from ice-sheet-wide velocity mapping, J. Glaciol., 56(197), 415–430, doi:10.3189/002214310792447734, 2010.

Joughin, I., Smith, B. E., Howat, I. M., Floricioiu, D., Alley, R. B., Truffer, M. and Fahnestock, M.: Seasonal to decadal scale variations in the surface velocity of Jakobshavn Isbrae, Greenland: Observation and model-based analysis, J. Geophys. Res. Earth Surf., 117(2), n/a-n/a, doi:10.1029/2011JF002110, 2012.

Joughin, I., I. Howat, B. Smith, and T. Scambos. MEaSUREs Greenland Ice Velocity: Selected Glacier Site Velocity Maps from InSAR, Version 1. Boulder, Colorado USA. NASA National Snow and Ice Data Center Distributed Active Archive Center, doi:http://dx.doi.org/10.5067/MEASURES/CRYOSPHERE/nsidc-0481.001, 2011, updated 2016.

Koenig, L. S., Lampkin, D. J., Montgomery, L. N., Hamilton, S. L., Turrin, J. B., Joseph, C. A., Moutsafa, S. E., Panzer, B., 5 Casey, K. A., Paden, J. D., Leuschen, C. and Gogineni, P.: Wintertime storage of water in buried supraglacial lakes across the Greenland Ice Sheet, Cryosphere, 9(4), 1333-1342, doi:10.5194/tc-9-1333-2015, 2015.

Krabill, W., Hanna, E., Huybrechts, P., Abdalati, W., Cappelen, J., Csatho, B., Frederick, E., Manizade, S., Martin, C., 10 Sonntag, J., Swift, R., Thomas, R. and Yungel, J.: Greenland Ice Sheet: Increased coastal thinning, Geophys. Res. Lett., 31(24), 1-4, doi:10.1029/2004GL021533, 2004.

Krawczynski, M. J., Behn, M. D., Das, S. B. and Joughin, I.: Constraints on the lake volume required for hydro-fracture through ice sheets, Geophys. Res. Lett., 36(10), L10501, doi:10.1029/2008GL036765, 2009.

15

Lampkin, D. J.: Supraglacial lake spatial structure in western Greenland during the 2007 ablation season, J. Geophys. Res. Earth Surf., 116(4), F04001, doi:10.1029/2010JF001725, 2011.

Lampkin, D. J. and VanderBerg, J.: A preliminary investigation of the influence of basal and surface topography on 20 supraglacial lake distribution near Jakobshavn Isbrae, western Greenland, Hydrol. Process., 25(21), 3347-3355, doi:10.1002/hyp.8170, 2011.

Lampkin, D. J., Amador, N., Parizek, B. R., Farness, K. and Jezek, K.: Drainage from water-filled crevasses along the margins of Jakobshavn Isbræ: A potential catalyst for catchment expansion, J. Geophys. Res. Earth Surf., 118(2), 795-813, doi:10.1002/jgrf.20039, 2013.

25

Luthcke, S. B., Zwally, H. J., Abdalati, W., Rowlands, D. D., Ray, R. D., Nerem, R. S., Lemoine, F. G., McCarthy, J. J. and Chinn, D. S.: Recent Greenland Ice Mass Loss by Drainage System from Satellite Gravity Observations, Science., 314(5803), 2006.

30

McMillan, M., Nienow, P., Shepherd, A., Benham, T. and Sole, A.: Seasonal evolution of supra-glacial lakes on the Greenland Ice Sheet., 2007.