# Peer review of "SPATIAL AND TEMPORAL VARIABILITY OF WATER-FILLED CREVASSE HYDROLOGIC STATES ALONG THE SHEAR MARGINS OF JAKOBSHAVN ISBRAE, GREENLAND"

_The Cryosphere, 2017_

## Referee Comment (RC1) · Anonymous Referee #1 · 3 Aug 2017

This paper explores potential relationships between the presence or absence of water-filled crevasse groups along the Jakobshavn glacier, and (1) air temperatures, (2) strain rates at the ice sheet surface, and (3) calving front position. No clear relationship is found with these three quantities, although hints of signals may be present that the authors pursue.

The novel contribution of this paper is, then, the dataset for the presence or absence of ponded water for the seven crevasse groups studied. This dataset was begun by Lampkin et al (2013) for the year 2007, and here has been extended from 2000 to 2015.

[Figure]

Indeed, the authors state that their objective is the collection of this binary dataset, which is the most comprehensive dataset available for this topic (Section 1.1). Yet the data are not fully presented. For instance, instead of plotting a time series of "full" / "drained" for each crevasse group, the authors plot histograms and box-and-whisker plots of monthly averages, which obscure the data by overly summarizing it. While this does not seem like an egregious violation of an openly-sharing data philosophy that The Cryosphere may have, it does feel like the authors are holding their cards unnecessarily close, which makes it difficult to engage with the material and evaluate the hypotheses presented. There is some value in histograms and statistical plots, but these should be presented alongside a full plot of the dataset, not instead of it.

For instance, the paper might be substantially improved with the addition of a few time series of "full" / "drained", with time series of the other variables (air temperature, strain rate, and calving front position) superimposed. This is approximately done in Figure 5, but because the data are limited to yearly totals or averages, a limited amount of information can be gleaned. Repeating the analyses in more detail and presenting them in full detail is a primary recommendation.

Another considerable shortcoming within this work is an immature treatment of event detection. At several points, the authors rightly declare the likely conflict between the sparsity of observations in the early part of the study (2008 and before) and the lower number of drainage events observed then. Figure 2a shows the considerable range in number observations over time, and this is valuable. Yet the authors still speculate on the possible causes (e.g. increased air temperatures, Section 4.3) of the apparent increase in drainage rate since 2006, and note the apparent decrease in the movement of the calving front over time (Section 4.5). This was irresponsible, given the limitations of the drainage dataset. I recommend assessing whether the apparent increase in the number of drainage events in recent years is real, following analytical techniques from any upper-level statistics text.

Finally, the recent paper by Everett et al (2016) is missing from this manuscript. That

work studied a very similar phenomenon on Helheim and was able to make a conclusion about what drives the drainage of water-filled crevasses on that glacier. Consideration and comparison of Jakobshavn to that system could add some good science here, but at the very least, needs to be included as it is the only other group, to my knowledge, studying this phenomenon.

For these reasons, I do not recommend publication at this time. With more work, the authors should be able to continue the analysis and complete their presentation of their dataset to create a future manuscript on this topic potentially worthy of publication in The Cryosphere.

Specific points

P1 L17 and elsewhere Strain rates of 1.2 /second are very high, more like a putty or a lava flow than a glacier. The correct unit is probably /year, this should be checked.

P3 L7 Google Earth is not a satellite

P3 L8 Some elaboration on how the 7 data sources "offset the relative performance limitations" of each other is required. As far as I can tell, it just results in a denser time series.

P3 L20 The NSIDC velocity dataset used here has approximately 11-day temporal resolution for Jakobshavn, yet only yearly strain rates are obtained and presented. This puzzled me greatly. Certainly much more can be learned with the level of detail available in this dataset. Why was the choice to analyze only on a yearly level made? This should be explained.

P4 L1 More detail is needed in the methods for identification. Is the method for detection of "filled" or "drained" automated or manual? What are the thresholds? Are any "in-between" states observed, and how would they be classified?

P4 L11 These data are posted at 100 meter resolution, yet the crevassed areas appear to be considerably larger than that. How are the strain rate data interpolated and/or

smoothed to account for this?

P5 L16 "multi-drain event" should be defined

P6 L28 How is the calving front tracked? (Data source, analysis techniques, presentation of data.) I was surprised to see a very smooth curve for calving front position Figure 6, as usually they are very jagged.

P7 L11 The discussion section should be better organized (it is currently one 60-line paragraph!) and extent the specific results into general conclusions. The literature review on hydrofracture does not belong here. This section was very difficult to follow and needs a lot of work.

Figure 1 Adding velocity contours or elevation contours would give a better sense of where the crevasse groups are located within the glacier system. The scale bar is too small and the color is very hard to read.

Figure 3 could be combined into Figure 2a, or better yet the y-axis here could be the percentage of time that a crevasse group was filled or drained.

Figure 4 Are the pattern groups identified here meaningful or discussed elsewhere in the manuscript?

Figure 5 This shows that 2012 was one of the coolest years on record. I am skeptical of this because 2012 is well known as a very big melt year.

Figure 6 Why is the calving front position so smooth? This cannot be correct (see comment above) and is not explained.

References

Everett et al, 2016. Annual down-glacier drainage of lakes and water-filled crevasses at Helheim Glacier, southeast Greenland. Journal of Geophysical Research

---

## Referee Comment (RC2) · Anonymous Referee #2 · 31 Aug 2017

This paper looks at the hydrology of the shear margins of Jakobshavn Isbrae, and attempts to link the filling and drainage of water-filled crevasses with the dynamics of the glacier both in cause and effect. This is a fascinating area of study; there are extensive regions of water-filled crevasses in Greenland yet they have received far less attention than supraglacial lakes in the literature, and their potential influence on the glacier's shear margins is intriguing and I can only commend the author's for seeking out an interesting topic of study. However, this paper falls short of its stated objectives and reading it left me no wiser as to the causes or effects of water-filled crevasse

drainage.

My major issues with it are as follows:

1) The authors make much of increases in drainage over the study period, yet don't take into account a similar trend in observation frequency. Any potential bias needs to be accounted for before any such conclusions can be reached.

2) I can't actually work out what the main conclusions of this paper are, let alone if they are supportable. Perhaps a separate conclusion section might have helped me understand what the point is? The discussion is based heavily on the existing literature and the conclusions hinted at appear to result from reviewing the literature rather than insight from the presented results. I don't believe that the results presented in the paper actually add anything to the discussion.

3) The paper is badly written with little evidence of care or proof reading. Some sections have incomplete sentences. Others are overly verbose. This does not help the reader fathom what conclusions they are expected to take home.

Unfortunately, I cannot recommend publication in its current state.

Line by line comments:

Page 1. 14: May has the fewest filled days? Not, for example, December?

15: "Inter-seasonal drain frequencies over this system...". I've read this several times and I'm not sure what it means.

17 (and throughout the paper): Do you mean averaged between -1 and 2? Or averaged from -1 to 2?

Page 2: 4: See Joughin et al. (1996) and Everett et al. (2016)

19: Is there a citation for these crevasse systems being responsible for hydrological weakening of the shear margin?

20-21: This is not a sentence.

26: "satellite derived measured velocity data". Redundant word

27: Again, it feels like the authors never finished this sentence.

Page 4: 18: Presumably you know how many crevasse groups were filled, making inferences from the number of drainages rather pointless?

Page 7 onwards: This whole section is very hard to read. Some paragraphs might help!

Figure 4a: No matter how much I read the description of this figure I cannot work out what it actually presents. I also don't see how it fits into the paper as a whole.

Everett, A., Murray, T., Selmes, N., Rutt, I., Luckman, A., James, T., Clason, C., O'Leary, M.,Karunarathna, H., Moloney, V. & Reeve, D. (2016). Annual down-glacier drainage of lakes and water-filled crevasses at Helheim Glacier, southeast Greenland. J. Geophys. Res. Earth Surf., 121, 1819–1833, doi:10.1002/2016JF003831.

Joughin, Ian and Tulaczyk, Slawek and Fahnestock, Mark and Kwok, Ron (1996) A Mini-Surge on the Ryder Glacier, Greenland, Observed by Satellite Radar Interferometry. Science, 274 (5285). pp. 228-230.

---

## Author Comment (AC1) · 14 Oct 2017

Comment #1 "... the data are not fully presented. For instance, instead of plotting a time series of "full" / "drained" for each crevasse group, the authors plot histograms and box-and-whisker plots of monthly averages, which obscure the data by overly summarizing it. There is some value in histograms and statistical plots, but these should be presented alongside a full plot of the dataset, not instead of it."

Reply #1 We are posting our dataset, along with a complete time series of the dataset

as supplemental material.

Comment #2 "For instance, the paper might be substantially improved with the addition of a few time series of "full" / "drained", with time series of the other variables (air temperature, strain rate, and calving front position) superimposed. This is approximately done in Figure 5, but because the data are limited to yearly totals or averages, a limited amount of information can be gleaned. Repeating the analyses in more detail and presenting them in full detail is a primary recommendation."

Reply #2 The time series will be presented as supplemental material, along with the dataset. The time series can be difficult for the reader to interpret with 7 crevasse groups presented over 16 years. Superimposing other variables make the plot even more difficult to interpret.

Comment #3 "Another considerable shortcoming within this work is an immature treatment of event detection... I recommend assessing whether the apparent increase in the number of drainage events in recent years is real, following analytical techniques from any upper-level statistics text."

Reply #3 We have performed a Spectral Decomposition along with a logit regression analysis. These sections have been appropriately added in the methods, results, and discussion.

P5 L7: 3.3 Spectral Decomposition of Unevenly Sampled Data P6 L1: 3.4 Logistic Regression Modeling P9 L10: 4.6 Hydrologic States Spectra P9 L15: 4.7 Logit Model P10 L6: 5.2 Power Spectra P10 L28-10: "This is also consistent in the logit model results as we might expect to see statistically significant trends in the probability of filled states with time commensurate with an increase in temperatures. The lack of significance in the trend on the probability of filled states regardless of the sign seems to indicate that temperature is not a control on whether a given crevasse group will be more or less likely to be filled."

Comment #4 "the recent paper by Everett et al (2016) is missing from this manuscript. That C2 TCD Interactive comment Printer-friendly version Discussion paper work studied a very similar phenomenon on Helheim and was able to make a conclusion about what drives the drainage of water-filled crevasses on that glacier. Consideration and comparison of Jakobshavn to that system could add some good science here, but at the very least, needs to be included as it is the only other group, to my knowledge, studying this phenomenon."

Reply #4 I have mentioned the hypothesis for crevasse drainage outlined in Everett et al., prior to factors we investigated that may influence crevasse drainage.

P10 L13-19: Everett et al., (2016) hypothesize that drainage and filling downstream of Helheim Glacier may be the result of a high pressure wave passing down glacier following a lake drainage. We have not observed coordination in drain and fill behaviors among adjacent pond groups. There is no relationship between supraglacial lake drainage and water-filled crevasse drainage within the shear margins of Jakobshavn as the closest lake to many of our CV groups is more than 15km away in the extra-marginal ice field. Lastly, it is not feasible for drainage of crevasse groups within the northern margin to impact the filling and drainage behavior of those within the southern margin and vice versa. The margins are separated by a deep trough with no evidence for connected subglacial hydrology transverse to the main direction of ice flow.

Comment #5 "P1 L17 and elsewhere Strain rates of 1.2 /second are very high, more like a putty or a lava flow than a glacier. The correct unit is probably /year, this should be checked."

Reply #5 The reviewer is correct, we had a unit error. The correct units were 1/annual. This has been changed throughout the paper, and in figure 5.

Comment #6 "P3 L7 Google Earth is not a satellite"

Reply #6 Deleted google earth from the list of satellites to emphasize that it is a platform

that combines imagery from 3 satellites.

Comment #7 "P3 L8 Some elaboration on how the 7 data sources "offset the relative performance limitations" of each other is required. As far as I can tell, it just results in a denser time series."

Reply #7 Our original wording is confusing. We have changed the wording to the reviewer's suggestion, which better represents the benefit to using multiple satellite platforms.

P3 L8-10: "Data from Landsat-7 ETM+, Landsat-8 OLI, Quickbird-1/2, Geo-Eye, Worldview-1/2, EO-1 ALI, SPOT-5 and ASTER, are used in this analysis. The combination of data from these systems increases the frequency of sampling resulting in enhanced temporal resolution which offsets the impact of cloud cover."

Comment #8 "P3 L20 The NSIDC velocity dataset used here has approximately 11-day temporal resolution for Jakobshavn, yet only yearly strain rates are obtained and presented. This puzzled me greatly. Certainly much more can be learned with the level of detail available in this dataset. Why was the choice to analyze only on a yearly level made? This should be explained."

Reply #8 It is true that velocities are available at higher temporal resolution but we opted to assess strain rates from yearly data in this analysis to provide a first order assessment of changes in annual strain rates at a temporal scale where our observation data set is most robust. We have inconsistent sampling from season to season and we took a conservative approach by not attempting to attribute inter-seasonal changes in strain rates to observed drainage events could be spurious. The scope of this paper was not to attribute each drainage event to a specific process as we assess that the sampling limitations inherent and acknowledge that our current data is not sufficient at this time to warrant this level of attribution. In future efforts, we intend to seek out additional data sets to establish a more seasonally consistent sampling interval over the archive.
Comment #9 "P4 L1 More detail is needed in the methods for identification. Is the method for detection of "filled" or "drained" automated or manual? What are the thresholds? Are any "in-between" states observed, and how would they be classified?"

Reply #9 The details addressing this question were clearly articulated in the manuscript (see page 4 L10-18). We have added additional content (page 4, line 18-19) to address your point about the inability to distinguish if a given pond was at its maximum or minimum extent when observed. We assume this is what you referring to as "in-between" state. If not, we already address that we are not able to determine any information about changes in the extent of the pond between successive images or the exact date of drainage if the pond was observed to contain water at the beginning of an observation interval and lacked water in the subsequent image.

P4 L10 "The hydrologic state of water-filled crevasse systems ($\psi$) are quantified through visual interpretation of imagery. P4 L18-19 "We did not document the areal extent of ponds and do not record partial drainage events. If water is present at all regardless of pond size, we designate the pond as "filled" otherwise it is classified as "drained"."

Comment #10 "P4 L11 These data are posted at 100 meter resolution, yet the crevassed areas appear to be considerably larger than that. How are the strain rate data interpolated and/or smoothed to account for this?"

Reply #10 The method for application of the strain data are described on P5 L2-6.

Comment #11 "'multi-drain event" should be defined' Reply #11 Multi-drain event is now defined prior to first use in the paper outside of the abstract.

P6 L20-21 "Some systems were observed to fill and drain more than once during a season, this will be referred to as a 'multi-drainage' event for the remainder of the paper. "

Comment #12 P6 L28 How is the calving front tracked? (Data source, analysis tech-

niques, presentation of data.) I was surprised to see a very smooth curve for calving front position Figure 6, as usually they are very jagged.

Reply #12 A data section has been added to the paper for our calving front data P4 L2-7. The data is acquired from the ESA archive without any further augmentation or processing.

Comment #13 P7 L11 The discussion section should be better organized (it is currently one 60-line paragraph!) and extent the specific results into general conclusions. The literature review on hydrofracture does not belong here. This section was very difficult to follow and needs a lot of work.

Reply #13 The literature review on hydrofracture has been moved to the motivation and prior work section on pages 1 and 2. We have also reorganized the discussion section to be easier to read.

Comment #14 "Figure 1 Adding velocity contours or elevation contours would give a better sense of where the crevasse groups are located within the glacier system. The scale bar is too small and the color is very hard to read."

Reply #14 We have added in elevation contours to figure 1 as recommended.

P20 L1: Figure 1: Study area showing the location of water-filled crevasse systems (CV) (white) within the shear margins of Jakobshavn Isbræ, west-central Greenland. The spatial extent is a composite based on observed areal extent from cloud-free, Landsat-7 panchromatic imagery only from 2000- 2013. Contours of elevation in meters are superimposed.

Comment #15 "Figure 3 could be combined into Figure 2a, or better yet the y-axis here could be the percentage of time that a crevasse group was filled or drained."

Reply #15 We believe figure 3 does offer some valuable information for this paper. We have opted to keep figure three the way it currently is. However, we have taken the advice and created a new figure (A2) that has the percent of filled days for the entire

study period for each CV group.

P30: Added Figure A2

Comment #16 "Figure 4 Are the pattern groups identified here meaningful or discussed elsewhere in the manuscript?"

Reply #16 The patterns are discussed on page 7, L22-24.

Comment #17 "Figure 5 This shows that 2012 was one of the coolest years on record. I am skeptical of this because 2012 is well known as a very big melt year"

Reply #17 Although peculiar, we double checked our temperature data from Swiss Camp and Jar-1, and the average displayed on the figure for 2012 is correct.

Comment #18 "Figure 6 Why is the calving front position so smooth? This cannot be correct (see comment above) and is not explained."

Reply #18 The calving front is plotted with the data from ESA archives as is. We have not manipulated or augmented the data in any way.

Please also note the supplement to this comment:
https://www.the-cryosphere-discuss.net/tc-2017-86/tc-2017-86-AC1-supplement.zip

---

## Author Comment (AC2) · 14 Oct 2017

Comment #1 "The authors make much of increases in drainage over the study period, yet don't take into account a similar trend in observation frequency. Any potential bias needs to be accounted for before any such conclusions can be reached."

Reply #1 The major bias that we observed was the inconsistent sampling due to cloud cover. We acknowledges this bias, which limited the scope to which we could attribute any one process to the observed behaviors of these systems.

[Figure]

Comment #2 "I can't actually work out what the main conclusions of this paper are, let alone if they are supportable. Perhaps a separate conclusion section might have helped me understand what the point is? The discussion is based heavily on the existing literature and the conclusions hinted at appear to result from reviewing the literature rather than insight from the presented results. I don't believe that the results presented in the paper actually add anything to the discussion." Reply #2 We have attempted to revise to make our conclusions clear. We separated the discussion and the conclusions section to P12 L3.

Comment #3 "The paper is badly written with little evidence of care or proof reading. Some sections have incomplete sentences. Others are overly verbose. This does not help the reader fathom what conclusions they are expected to take home."

Reply #3 We have substantially revised the paper to address grammatical, textual, and structural errors.

Comment #4 "Page 1. 14: May has the fewest filled days? Not, for example, December?"

Reply #4 The study period is from 2000-2015 for the months of May-September. I have revised this to make it clear in the abstract before the mention of May having the fewest filled days.

P1 L11-12: "A fusion of multi-sensor optical satellite imagery was used to examine hydrologic states during the melt season (May to September) from 2000 to 2015."

Comment #5 "Inter-seasonal drain frequencies over this system...". "I've read this several times and I'm not sure what it means."

Reply #5 This means that throughout a given melt season, the amount of drainage could vary between 0-5. I have reworded this section to make it clearer.

P1 L14: "The number of drainages per crevasse group in a season ranged from 0 to 5."

Comment #6 "(and throughout the paper): Do you mean averaged between -1 and 2? Or averaged from -1 to 2?"

Reply #6 We mean that for any given melt season in the study period, the average temperature ranged between -1 and 2C.

P1 L16: "Over the study period, average summer temperatures ranged from -1 and 2 âĄřC and..."

Comment #7 "Page 2: 4: See Joughin et al. (1996) and Everett et al. (2016)"

Reply #7 I have added in the suggested citation.

P1 L24-28; P2 L1 "Commensurate with these changes has been the documented impact of surface meltwater on ice sheet velocity during the summer within the ablation zone (Zwally et al., 2002; Joughin et al., 2008; van de Wal et al., 2008; Shepherd et al., 2009; Bartholomew et al., 2010; Sundal et al., 2011; Palmer et al., 2011; Hoffman et al., 2011), via supraglacial lakes, channels, and moulins largely beyond regions of fast flow (Echelmeyer et. al., 1991; Joughin et al., 1996; Box and Ski, 2007; McMillan et al., 2007; Sneed and Hamilton, 2007; Das et al., 2008, Sundal et al, 2009; Lampkin, 2011; Selmes et al., 2011; Tedesco and Steiner, 2011; Howat et al., 2013; Koenig et al., 2015).

Comment #8 "19: Is there a citation for these crevasse systems being responsible for hydrological weakening of the shear margin?"

Reply #8 The impact of crevasse drainage weakening the shear margins is noted in Lampkin et al., 2013. I have added the citation to the manuscript.

P2 L21-22: "We characterize temporal patterns in the drain state of water-filled crevasse systems responsible for hydrologic weakening of Jakobshavn Isbrae (Lampkin et al., 2013)."

Comment #9 "20-21: This is not a sentence."

Reply #9 Added in "is evaluated" to complete the sentence

P2 L23-24: "This work provides an important benchmark from which future changes in this component of supraglacial hydrology and the role of meltwater in fast flowing ice streams are evaluated."

Comment #10 "satellite derived measured velocity data". Redundant word

Reply #10 Removed the word 'measured' due to redundancy

P2 L29-30: "We examine spatial and temporal variability of surface strain fields based on satellite-derived velocity data."

Comment #11 "27: Again, it feels like the authors never finished this sentence."

Reply #11 Added 'data are used' to make the sentence a complete thought.

P2 L30 P3 L1-2: "Additionally, near surface air temperature data are used to evaluate meteorological conditions associated with melt production and runoff that can influence the hydrologic state of the water-filled crevasse systems."

Comment #12 "Page 4: 18: Presumably you know how many crevasse groups were filled, making inferences from the number of drainages rather pointless?"

Reply #12 The paper explores the frequency of drainages in a given season for each crevasse group. If a crevasse can fill and drain more than one time, then presumably more meltwater can be injected to the bedrock, and enhance ice loss.

Comment #13 "Page 7 onwards: This whole section is very hard to read. Some paragraphs might help!"

Reply #13 We have separated the sections, and revised them to be more clear.

Comment #14 "Figure 4a: No matter how much I read the description of this figure I cannot work out what it actually presents. I also don't see how it fits into the paper as a whole."

Reply #14 This figure shows for each crevasse group, how many days it was filled each month for each year. This gives us a visual representation of the filling and drainage patterns. Figure b shows over the entire study period the mean amount of days that each group was designated the filled state. The axis on fig 4a has been adjusted to be more easily read.

P23 L1: Adjusted figure.

Please also note the supplement to this comment:
https://www.the-cryosphere-discuss.net/tc-2017-86/tc-2017-86-AC2-supplement.zip